# Entanglement-Structured LSTM Boosts Chaotic Time Series Forecasting

**DOI:** 10.3390/e23111491

**Published:** 2021-11-11

**Authors:** Xiangyi Meng, Tong Yang

**Affiliations:** 1Center for Complex Network Research and Department of Physics, Northeastern University, Boston, MA 02115, USA; x.meng@neu.edu or; 2Department of Physics, Boston University, Boston, MA 02215, USA; 3Department of Physics, Boston College, Chestnut Hill, MA 02467, USA

**Keywords:** quantum entanglement, recurrent neural networks, tensorization, chaotic dynamical system, chaotic time series forecasting

## Abstract

Traditional machine-learning methods are inefficient in capturing chaos in nonlinear dynamical systems, especially when the time difference Δt between consecutive steps is so large that the extracted time series looks apparently random. Here, we introduce a new long-short-term-memory (LSTM)-based recurrent architecture by tensorizing the cell-state-to-state propagation therein, maintaining the long-term memory feature of LSTM, while simultaneously enhancing the learning of short-term nonlinear complexity. We stress that the global minima of training can be most efficiently reached by our tensor structure where all nonlinear terms, up to some polynomial order, are treated explicitly and weighted equally. The efficiency and generality of our architecture are systematically investigated and tested through theoretical analysis and experimental examinations. In our design, we have explicitly used two different many-body entanglement structures—matrix product states (MPS) and the multiscale entanglement renormalization ansatz (MERA)—as physics-inspired tensor decomposition techniques, from which we find that MERA generally performs better than MPS, hence conjecturing that the learnability of chaos is determined not only by the number of free parameters but also the tensor complexity—recognized as how entanglement entropy scales with varying matricization of the tensor.

## 1. Introduction

Time series forecasting [1], despite its undoubtedly tremendous potential in both theoretical issues (e.g., mechanical analysis, ergodicity) and real-world applications [2] (e.g., traffic, weather, and clinical records analysis), has long been known as an intricate field. From classical work on statistics such as auto-regressive moving average (ARMA) families [3] and basic hidden Markov models (HMM) [4,5] to contemporary machine-learning (ML) methods [6,7,8,9] such as gradient boosted trees (GBT) and neural networks (NN), the essential complexity in time series has been more and more frequently recognized. In particular, forecasting models have extended their applicable range from linear, Markovian cases to nonlinear, non-Markovian, and even more general situations [10]. Among all known methods, recurrent NN architectures [11], including plain recurrent neural networks (RNN) [12] and long short-term memory (LSTM) [13], are the most capable of capturing this complexity, as they admit the fundamental recurrent behavior of time series data. LSTM has proved useful in speech recognition and video analysis tasks [14] in which maintaining long-term memory is essential to the complexity. In relation to this objective, novel architectures such as higher-order RNN/LSTM (HO-RNN/LSTM) [15] have been introduced to capture long-term non-Markovianity explicitly, further improving performance and leading to more accurate theoretical analysis.

Still, another domain of complexity—chaos—has been far less understood [16,17]. Even though enormous theory/data-driven studies on forecasting chaotic time series by means of recurrent NN have been conducted [18,19,20,21,22,23,24,25], there is still a lack of consensus on which features play the most important roles in the forecasting methods. The notorious indication of chaos,
(1)$$\begin{array}{c} \left|\delta x_{t}\right|\approx e^{\lambda t}\left|\delta x_{0}\right| \end{array}$$
(where λ denotes the spectrum of Lyapunov exponents), suggests that the difficulty of forecasting chaotic time series is two-fold: first, any small error will propagate exponentially, and thus multi-step-ahead predictions will be exponentially worse than one-step-ahead ones; second, and more subtly, when the actual time difference Δt between consecutive steps increases, the minimum redundancy of model capacity needed for smoothly descending to the global minima (or sufficiently good local ones) during NN training also increases exponentially. Most studies only address the first difficulty by improving the prediction accuracy achievable at the global minima. Yet the latter is in fact more crucial, especially when Δt is so large that the time series looks apparently random and a trivial local minimum would most likely be reached instead. Recently, tensorization has been introduced in recurrent NN architectures [26,27]. A tensorized version of HO-RNN/LSTM, namely, HOT-RNN/LSTM [28], has claimed an advantage in learning long-term nonlinearity in Lorenz systems of small Δt. On the one hand, we believe that the global minima of chaos (where the dominance of linear dependence is absent) can be most efficiently reached through tensorization approaches, where all nonlinear terms, up to some polynomial order, are treated explicitly and weighted equally. On the other hand, for simple chaotic dynamical systems, nonlinear complexity is only encoded in the short term, not the long term, which HO/HOT models will not be efficient in capturing when Δt is large. Hence, a new tensorization-based recurrent NN architecture is desired so as to foster our understanding of chaos in time series and to meet practical needs, e.g., the modeling of laminar flame fronts and chemical oscillations [29,30,31,32].

In this paper, we introduce a new LSTM-based architecture by tensorizing the cell-state-to-state propagation therein, retaining the long-term memory features of LSTM while simultaneously enhancing the learning of short-term nonlinear complexity. Compared with traditional LSTM architectures, including stacked LSTM [33] and other aforementioned statistics/ML-based forecasting methods, our model is shown to be a general and outperforming approach for capturing chaos in almost every typical chaotic continuous-time dynamical system and discrete-time map with controlled comparable NN training conditions, justified by both our theoretical analysis and experimental results. Our model is also tested on real-world time series datasets, where the improvements range up to 6.3%.

During the tensorization, we have explicitly embedded many-body quantum state structures—a way of reducing the exponentially large degree of freedom of a tensor (i.e., tensor decomposition)—popularly studied in condensed matter physics, which is not unseen in NN design [34]. A many-body entangled state living in a tensor-product Hilbert space is hardly separable. The same inseparability behavior also appears in nonlinear multi-variate functions when crossing terms between different variables become too complex. This similarity motivated us to adopt a special measure of tensor complexity, namely, the entanglement entropy (EE) [35], which is commonly used in quantum physics and quantum information [34]. For one-dimensional many-body states, two thoroughly studied, popular but different structures exist—multiscale entanglement renormalization ansatz (MERA) [36] and matrix product states (MPS) [37], of which the EE scales with the subsystem size or not at all, respectively [35]. For most pertinent studies, MPS has been proven to be efficient enough to be applicable to a variety of tasks [38,39,40,41]. However, our experiments show that, regarding our entanglement-structured design of the new tensorized LSTM architecture, LSTM-MERA performs even better than LSTM-MPS in general without increasing the number of parameters. This finding leads to another interesting result. We conjecture that not only should tensorization be introduced, but the tensor’s EE has to scale with the system size as well; hence, MERA is more efficient than MPS in learning chaos.

## 2. Recurrent Architecture and Tensorization

### 2.1. Formalism of LSTM Architecture

The formalism starts from an operator-theoretical perspective by defining two general types of real *operators*, W and σ, through which most NN architectures can be represented. W:X→G is simply a linear operator, but σ:G→G is a nonlinear operator that σ(G)=(σ∘G)∈G given G∈G where ∘ stands for the entry-wise operator product. All double-struck symbols (X,G,⋯) used in this context are general real vector spaces considered to be of *covariant* type, as W can be interpreted as a linear-map-induced 2-*contravariant* bilinear form. Next, a *state propagation function* (i.e,. a gate) g(x,y,⋯;W)=σ(W(x⊕y⊕⋯)) is introduced, where x⊕y⊕⋯ stands for the tensor direct sum of real vectors x,y,⋯. Following the formalism, an LSTM architecture can be expressed as follows:(2)$$\begin{array}{ccc} s_{t} & = & g\left(1,x_{t-1},s_{t-1};W_{o}\right)\circ \sigma \left(c_{t}\right),\;x_{t}=g\left(1,s_{t};W_{x}\right),\\ c_{t} & = & g\left(1,x_{t-1},s_{t-1};W_{f}\right)\circ c_{t-1}\\ & + & g\left(1,x_{t-1},s_{t-1};W_{i}\right)\circ g\left(1,x_{t-1},s_{t-1};W_{m}\right), \end{array}$$
where the four gates controlled by $W_{i}$, $W_{m}$, $W_{f}$, and $W_{o}$ are the input, memory, forget, and output gates. The state $s_{t}$ and the cell state $c_{t}$ are *h*-dimensional *covectors*, whereas the input $x_{t}$ is a *d*-dimensional covector (Figure 1). Therefore, $W_{i}$ (as well as $W_{m}$, $W_{f}$, and $W_{o}$) has a direct-sum contravariant realization as a matrix $W_{i}\in M\left(h,1\right)⊕M\left(h,d\right)⊕M\left(h,h\right)$ that contains h(1+d+h) free real parameters at most. During NN training, only these free parameters of the linear operators are learnable, whereas all σ (i.e., activation functions) are fixed to be tanh, sigmoid, or other nonlinear functions. The cell state $c_{t}$ is designed to suffer less from the vanishing gradient problem and thus to capture long-term memory better, whereas $s_{t}$ tends to capture short-term dependence.

### 2.2. Tensorized State Propagation

Our tensorized LSTM architecture (Figure 1) is exactly based on Equation (Equation 2), from which the only change is:(3)$$s_{t}=g\left(1,x_{t-1},s_{t-1};W_{o}\right)\circ g\left(T\left(\sigma \left(c_{t}\right)\right);W_{T}\right).$$
$g(T\left(\sigma \left(c_{t}\right)\right);W_{T})$ is coined a *tensorized state propagation function*, for which $W_{T}:T\to G$ acts on as a covariant tensor
(4)$$T\left(\sigma \left(c_{t}\right)\right)=\underset{l}{⨂}\;\left(1⊕q_{t,l}\right)=\underset{l}{⨂}\;\left(1⊕W_{l}\left(\sigma \left(c_{t}\right)\right)\right).$$
Each $W_{l}$ in Equation (Equation 4) maps from $\sigma (c_{t})$ to a new covector $q_{t,l}\in Q$. Here, Q is named a *local q-space*, as, by way of analogy, considered as encoding the *local* degree of freedom in quantum mechanics. Q can be extended to the complex number field if necessary. Mathematically, Equation (Equation 4) offers the possibility of directly constructing orthogonal polynomials up to order *L* from $\sigma (c_{t})$ to build up nonlinear complexity. In fact, when *L* goes to infinity, $T=\lim_{L\to \infty }{\left(1⊕Q\right)}^{⊗L}=1⊕Q⊕Q⊗Q⊕\cdots$ becomes a tensor algebra (up to a multiplicative coefficient), and $T(\sigma \left(c_{t}\right))$ admits any nonlinear smooth function of $\sigma (c_{t})$.

We now realize Equation (Equation 4) by choosing *L* independent realizations, $W_{l}\in M\left(P-1,h\right)$, l=1,2,⋯,L, which in total contain L(P−1)h learnable parameters at most, each mapping $\sigma \left(c_{t}\right)\equiv \tanh c_{t}$ to a (P−1)-dimensional covector $q_{t,l}$,
(5)$$\left(\begin{array}{c} {\left[\tanh c_{t}\right]}_{1}\\ {\left[\tanh c_{t}\right]}_{2}\\ \vdots \\ {\left[\tanh c_{t}\right]}_{h} \end{array}\right)\to \left(\begin{array}{cccc} \left(\begin{array}{c} 1\\ {\left[q_{t}\right]}_{21}\\ {\left[q_{t}\right]}_{31}\\ \vdots \\ {\left[q_{t}\right]}_{P1} \end{array}\right)\; & \left(\begin{array}{c} 1\\ {\left[q_{t}\right]}_{22}\\ {\left[q_{t}\right]}_{32}\\ \vdots \\ {\left[q_{t}\right]}_{P2} \end{array}\right)\; & \begin{array}{c} \cdots \\ \cdots \\ \cdots \\ \ddots \\ \cdots \end{array}\; & \left(\begin{array}{c} 1\\ {\left[q_{t}\right]}_{2L}\\ {\left[q_{t}\right]}_{3L}\\ \vdots \\ {\left[q_{t}\right]}_{PL} \end{array}\right) \end{array}\right).$$
Following Equation (Equation 4), $T(\tanh c_{t})$ is simply the tensor product of all column vectors on the right hand side of Equation (Equation 5).

From the realization of Equation (Equation 3), $W_{T}\in M\left(h,P^{L}\right)$ [Figure 2a], however, a problem of exponential explosion (a.k.a. the “curse of dimensionality”) arises. Treating $W_{T}$ maximally by training all $hP^{L}$ learnable parameters is very computationally expensive, especially as *L* cannot be small because it governs the nonlinear complexity. To overcome this “curse of dimensionality”, *tensor decomposition* techniques have to be exploited [39] for the purpose of finding a much smaller subset $T\subset M(h,P^{L})$ to which all possible $W_{T}$ belong, without sacrificing any expressive power.

### 2.3. Many-Body Entanglement Structures

Below, we introduce the two many-body quantum state structures (MPS and MERA) as efficient low-order tensor decomposition techniques for representing $W_{T}$.

#### 2.3.1. MPS

As one of the most commonly used tensor decomposition techniques, MPS is also widely known as tensor-train decomposition [42] and takes the following form [Figure 2b]
$${\left[W_{T}\right]}_{\mu_{1}\cdots \mu_{L}}^{h}=\sum_{\left\{\alpha \right\}}^{D_{\mathrm{II}}}{\left[w_{0}\right]}_{\alpha_{1}\alpha_{L+1}}^{h}\left({\left[w_{1}^{†}\right]}_{\mu_{1}}^{\alpha_{1}\alpha_{2}}{\left[w_{2}^{†}\right]}_{\mu_{2}}^{\alpha_{2}\alpha_{3}}\cdots {\left[w_{L}^{†}\right]}_{\mu_{L}}^{\alpha_{L}\alpha_{L+1}}\right)$$
in our model, where $w_{1}^{†},w_{2}^{†},\cdots ,w_{L}^{†}$ are learnable 3-tensors (the symbol ${}^{†}$ denoting that they are *inverse isometries* [35]). $D_{\mathrm{II}}$ is an artificial dimension (the same for all α). $w_{0}$ is no more than a linear transformation that collects the boundary terms and maintains symmetry. The above notations are used for consistency with quantum theory [35] and the following MERA representation.

#### 2.3.2. MERA

The best way to explain MERA is using graphical tools, e.g., tensor networks [35]. MERA differs from MPS in its hierarchical tree structure: within each level {I,II,⋯}, the structure contains a layer of 4-tensor *disentanglers* of dimensions $\left\{D_{I}^{4},D_{\mathrm{II}}^{4},\cdots \right\}$, and then a layer of 3-tensor *isometries* of dimensions $\left\{D_{I}^{2}\times D_{\mathrm{II}},D_{\mathrm{II}}^{2}\times D_{\mathrm{III}},\cdots \right\}$, of which details can be found in [36]. MERA is similar to the Tucker decomposition [43] but fundamentally different because of the existence of disentanglers, which smear the inhomogeneity of different tensor entries [36].

Figure 2c shows the reorganized version of MERA used in our model, where the storage of independent tensors is maximally compressed before they are multiplied with each other by tensor products, which allows more GPU acceleration during NN training.

#### 2.3.3. Scaling Behavior of EE

Now we take advantage of an important measure of tensor complexity: the *entanglement entropy* (EE). Given an arbitrary tensor $W_{\mu_{1}\cdots \mu_{L}}$ of dimension $P^{L}$ and a cut *l* so that 1≤l≪L, the EE is defined in terms of the α-Rényi entropy [35],
(6)$$S_{\alpha }\left(l\right)\equiv S_{\alpha }\left(W\left(l\right)\right)=\frac{1}{1-\alpha }\log\frac{\sum_{i=1}^{P^{l}}\sigma_{i}^{\alpha }\left(W\left(l\right)\right)}{{\left(\sum_{i=1}^{P^{l}}\sigma_{i}\left(W\left(l\right)\right)\right)}^{\alpha }},$$
assuming α≥1. The Shannon entropy is recovered under α→1. $\sigma_{i}\left(W\left(l\right)\right)$ in Equation (Equation 6) is the *i*-th singular value of matrix $W\left(l\right)=W_{\left(\mu_{1}\times \cdots \times \mu_{l}\right),\left(\mu_{l+1}\times \cdots \times \mu_{L}\right)}$, matricized from $W_{\mu_{1}\cdots \mu_{L}}$. How S(l) scales with *l* determines how much redundancy exists in $W_{\mu_{1}\cdots \mu_{L}}$, which in turn reveals how efficient at most a tensor decomposition technique can be. For one-dimensional gapped low-energy quantum states (ground states), their EE is saturated even as *l* increases, i.e., $S_{\alpha }\left(l\right)=\Theta \left(1\right)$. Thus, their low-entanglement characteristics can be efficiently represented via MPS, of which the EE does not scale with *l* either and is bounded by $S_{\alpha }\left(l\right)\leq S_{1}\left(l\right)\leq 2\log D_{\mathrm{II}}$ [35]. By contrast, a non-trivial scaling behavior $S_{\alpha }\left(l\right)=\Theta \left(\log l\right)$ corresponds to gapless low-energy states and can only be efficiently represented by MERA, of which $S_{\alpha }\left(l\right)\leq S_{1}\left(l\right)\leq C+\sum_{\mathrm{level}=1}^{\log_{2}l}\log D_{\mathrm{level}}\approx C+C^{\prime }\log l$ scales logarithmically [36]. The bounds of both MPS and MERA have also been proven to be tight [35,36].

The different EE scaling behaviors of MERA and MPS have hence provided an apparent geometric advantage of MERA, i.e., its quasi-two-dimensional structure [Figure 2c], enlarging which will increase not only the width but also the *depth* of NN as the number of applicable levels scales logarithmically with *L*, offering even more power for model generalization on the already-inherited LSTM architecture [11]. Such an advantage is further confirmed by Equation (Equation 9) and then in Section 4.1, in which tensorized LSTMs with the two different representations LSTM-MPS and LSTM-MERA are tested.

## 3. Theoretical Analysis

### 3.1. Expressive Power

First, we prove the following theorem that links the variations of $c_{t}$ and $s_{t}$:

**Theorem** **1.**
*Given an LSTM architecture [Equation (Equation 2)], to which the input is a chaotic dynamical system $x_{t}$, characterized by a matrix λ of which the spectrum is the Lyapunov exponent(s), so that any variation $\delta x_{t}$ propagates exponentially [Equation (Equation 1)], then, up to the first order (i.e., $\delta x_{t-1}$),*

(7)
$$\begin{array}{c} \left|\delta s_{t}\right|\geq Ce^{\lambda }\left|\delta c_{t}\right|, \end{array}$$

*where $C\propto 1/{\left\|W\right\|}_{\infty }^{2}$ and ${\left\|\cdot \right\|}_{p=\infty }$ is the operator norm.*


**Proof.** From Equation (Equation 2), one has $\delta x_{t}=\left(\partial g\left(1,s_{t};W_{x}\right)/\partial s_{t}\right)\delta s_{t}$ where the first-order derivative is bounded by $\|\partial g\left(1,s_{t};W_{x}\right)/\partial s_{t}{\|}_{\infty }\leq \|W_{x}{\|}_{\infty }\|\sigma^{\prime }{\|}_{L_{\mu }^{\infty }}\leq {\left\|W_{x}\right\|}_{\infty }$, since the derivative of the active function supported on (−∞,∞) satisfies $\|\sigma^{\prime }{\|}_{L_{\mu }^{\infty }}\equiv {\left\|1/\cosh^{2}\right\|}_{L_{\mu }^{\infty }}\leq 1$. On the other hand, one has
$$\begin{array}{c} \delta c_{t}=\left[c_{t-1}\circ \partial g\left(1,x_{t-1},s_{t-1};W_{f}\right)/\partial x_{t-1}+\partial \left(g(1,x_{t-1},s_{t-1};W_{i})\right.\right.\\ \left.\left.\circ \;g(1,x_{t-1},s_{t-1};W_{m})\right)/\partial x_{t-1}\right]\delta x_{t-1}+O\left(\delta x_{t-2}\right)+\cdots ,\; \end{array}$$
and thus
$$|\delta c_{t}\left|\leq \right\|W_{f}{\|}_{\infty }\left|c_{t-1}\right|\circ \delta |x_{t-1}|+\left(\|W_{i}{\|}_{\infty }+{\left\|W_{m}\right\|}_{\infty }\right)\delta \left|x_{t-1}\right|,$$
which yields Equation (Equation 7), where w.l.o.g. all linear maps are assumed to be around the same magnitude $∼{\left\|W\right\|}_{\infty }$. Note that $\left|c_{t-1}\right|$ is also bounded because $|g(1,x_{t-1},s_{t-1};W_{f})|\leq 1$, which means $c_{t}$ is stationary. □

Equation (Equation 7) suggests that the state propagation from $c_{t}$ to $s_{t}$ carries the chaotic behavior. In fact, to preserve the long-term memorization in LSTM, $c_{t}$ has to depend on $c_{t-1}$ with a linear behavior and thus cannot carry chaos itself. This is further experimentally verified in Section 4.3. Nevertheless, note that Equation (Equation 1) is a necessary condition of chaos, not a sufficient condition.

Now, we look into the expressive power of our introduced tensorized state propagation function, $W_{T}T\left(\sigma \left(c_{t}\right)\right)$. One of the advantages of tensorizing the state propagation function in the form of Equation (Equation 4) is the well-behaved polynomial space constructed by the tensor product, by virtue of which the approximation of $W_{T}T$ to any (*k*-Sobolev) function *f* can always be bounded, as proven by the following theorem:

**Theorem** **2.**
*Let $f\in H_{\mu }^{k}\left(\Lambda \right)$ be a target function living in the k-Sobolev space, $H_{\mu }^{k}\left(\Lambda \right)=\left\{f\in L_{\mu }^{2}\left(\Lambda \right)\right|\left.\sum_{|i|\leq k}\|\partial^{\left(i\right)}{f\|}_{L_{\mu }^{2}\left(\Lambda \right)}<\infty \right\}$, where $\partial^{\left(i\right)}f\in L_{\mu }^{2}\left(\Lambda \right)$ is the i-th weak derivative of f, up to order k≥0, square-integrable on support $\Lambda ={\left(-1,1\right)}^{h}$ with measure μ. $W_{T}T\left(\sigma \left(c_{t}\right)\right)$ can approximate $f(\sigma \left(c_{t}\right))$ with an $L_{\mu }^{2}\left(\Lambda \right)$ error at most*

(8)
$$\|f-W_{T}{T\|}_{L_{\mu }^{2}\left(\Lambda \right)}\leq C\min{\left(L,\left\lfloor L\left(P-1\right)/h\right\rfloor \right)}^{-k}{\left\|f\right\|}_{H_{\mu }^{k}\left(\Lambda \right)}$$

*provided that $\left(h-1\right)hP^{L}\geq \left(h^{1+\min(L,\lfloor L(P-1)/h\rfloor )}-1\right)$. ${\left\|f\right\|}_{H_{\mu }^{k}\left(\Lambda \right)}=\sum_{|i|\leq k}\|\partial^{\left(i\right)}{f\|}_{L_{\mu }^{2}\left(\Lambda \right)}$ is the Sobolev norm and C is a finite constant.*


**Proof.** The *Hölder-continuous spectral convergence theorem* [44] states that $\|f-P_{N}{f\|}_{L_{\mu }^{2}\left(\Lambda \right)}\leq CN^{-k}{\left\|f\right\|}_{H_{\mu }^{k}\left(\Lambda \right)}$, in which $P_{N}:L_{\mu }^{2}\left(\Lambda \right)\to P_{N}$ is an orthogonal projection that maps *f* to $P_{N}f$. σ(c(t))∈Λ is guaranteed as σ≡tanh. The Sobolev space $P_{N}\subset L_{\mu }^{2}\left(\Lambda \right)$ is spanned by polynomials of a degree of at most *N*. Next, note that in the realization of $T(\sigma \left(c_{t}\right))$ each $W_{l}$ is independent [Equation (Equation 4)], and thus $P_{N}=\mathrm{span}\left(T\left(\sigma \left(c_{t}\right)\right)\right)$ is possible, where *N* is determined by *L*, *P*, and *h*. When P−1≥h, the maximum polynomial order is guaranteed *L*; when P−1<h, dim{Q}<dim{G}, and hence $T(\sigma \left(c_{t}\right))$ can only fully cover a polynomial order of up to ⌊L(P−1)/h⌋. Finally, Equation (Equation 8) is proven based on the fact that $W_{T}T$ maximally admits $P_{N}f$ as long as $hP^{L}\geq \sum_{i=0}^{N}h^{i}=\left(h^{N+1}-1\right)/\left(h-1\right)$, the latter of which is the size of the maximum orthogonal polynomial basis admitted by $P_{N}$. □

Equation (Equation 8) can be used to estimate how *L* scales with the chaos the dynamical system possesses. In particular, Equation (Equation 7) suggests that $\partial^{\left(1\right)}f∼e^{\lambda \Delta t}$, where Δt is the actual time difference between consecutive steps. Therefore, to persevere the error bound [Equation (Equation 8)] one at least expects that $L^{-1}∼e^{-\lambda \Delta t}$, i.e., *L* has to increase exponentially with respect to λΔt. To achieve this, tensorization is undoubtedly the most efficient approach, especially when Δt is large.

### 3.2. Worst-Case Bound by EE

The above analysis offers an intuitive bound on the expressive power of $W_{T}T$. Unfortunately, Equation (Equation 8) is valid only when all $hP^{L}$ degrees of freedom of $W_{T}$ are independent. A low-order tensor decomposition may therefore impact the expressive power.

Below, we compare the two different entanglement structures, MPS and MERA, which have major differences in their EE scaling behaviors. We proceed via the following theorem, relating the tensor approximation error to entanglement scaling:

**Theorem** **3.**
*Given a tensor ${\left[W_{T}\right]}_{\mu_{1}\cdots \mu_{L}}$ and its tensor decomposition ${\bar{W}}_{T}$, the worst-case p-norm (p≥1) approximation error is bounded from below by*

(9)
$$\begin{array}{cc} \; & \min_{\left\{{\bar{W}}_{T}\right\}}\|W_{T}-{\bar{W}}_{T}{\|}_{p}=\min_{\left\{{\bar{W}}_{T}\right\}}\max_{l\geq 1}{\left\|W_{T}\left(l\right)-{\bar{W}}_{T}\left(l\right)\right\|}_{p}\\ \geq & \min_{\left\{{\bar{W}}_{T}\right\}}\max_{l\geq 1}\left|e^{\frac{1-p}{p}S_{p}\left(W_{T}\left(l\right)\right)}\|W_{T}{\|}_{1}-e^{\frac{1-p}{p}S_{p}\left({\bar{W}}_{T}\left(l\right)\right)}{\left\|{\bar{W}}_{T}\right\|}_{1}\right|, \end{array}$$

*where $S_{\alpha \equiv p}\left(W\left(l\right)\right)$ is the α-Rényi entropy [Equation (Equation 6)].*


**Proof.** Equation (Equation 9) is easily proven by noting the Minkowski inequality ${\left\|A+B\right\|}_{p}\leq {\left\|A\right\|}_{p}+{\left\|B\right\|}_{p}$ and that $\left(1-\alpha \right)S_{\alpha }\left(l\right)=\alpha \log\|W_{T}{\left(l\right)\|}_{\alpha }-\alpha \log{\left\|W_{T}\left(l\right)\right\|}_{1}$ when α≡p≥1 [Equation (Equation 6)]. □

The worst-case bound [Equation (Equation 9)] is optimized whenever $S_{p}\left({\bar{W}}_{T}\left(l\right)\right)$ scales the same way as $S_{p}\left(W_{T}\left(l\right)\right)$ does. Assuming $S_{p}\left(W_{T}\left(l\right)\right)=C+C^{\prime }\log l$, then an MPS-type ${\bar{W}}_{T}$ cannot efficiently approximate $W_{T}$ unless $D_{\mathrm{II}}$ increases with logl too, from which the total number of free parameters $∼PLD_{\mathrm{II}}^{2}$ [Figure 2b], however, becomes unbounded. By contrast, a MERA-type ${\bar{W}}_{T}$ matches the scaling, by which the total number of free parameters $∼(D^{4}+D^{3})L$ (where $D\equiv D_{\mathrm{II}},\cdots =\exp C^{\prime }$) is efficient enough for any worst-case *l*.

It is unknown how quantitatively the failure to approximate $W_{T}$ may impact the expressive power given in Equation (Equation 8). The disappearance of the worst-case bound in Equation (Equation 9) is a necessary condition for Equation (Equation 8) to be valid.

## 4. Results

We investigated the accuracy of LSTM-MERA and its generalization ability on different chaotic time series datasets by evaluating the root mean squared error (RMSE) of its one-step-ahead predictions against target values. The benchmark for comparison was chosen to be a vanilla LSTM, of which the hidden dimension *h* was arbitrarily chosen in advance. LSTM-MERA (and other architectures if present) was built upon the benchmark.

Each time series dataset for training/testing consisted of a set of $N_{X}$ time series, $\left\{X^{i}|i=1,2,\cdots ,N_{X}\right\}$. Each time series $X^{i}=\left\{x_{t}^{i}|t\in T^{i}\right\}$ is of fixed length $|T^{i}|=\mathrm{input}\;\mathrm{steps}\;+1$ so that all but the last step of $X^{i}$ were inputs, whereas the last step was the one-step-ahead target to be predicted. The dataset $\left\{X^{i}\right\}$ was divided into two subsets—one for testing and one for training, which was further randomly split into a plain training set and a validation set by 80%:20%. Complete details are given in Appendix C.

All models were trained using Mathematica 12.0 on its NN infrastructure, Apache MXNet, using an ADAM optimizer with $\beta_{1}=0.9$, $\beta_{2}=0.999$, and $\epsilon =10^{-5}$. The learning rate $=10^{-2}$ and batch size =64 were chosen *a priori*. The NN parameters producing the lowest validation loss during the entire training process were accepted.

### 4.1. Comparison of LSTM-Based Architectures

When evaluating the advantages of LSTM-MERA, a controlled comparison is essential to confirm that the architecture of LSTM-MERA is inherently better than that of other architectures, not just because the increase of the number of free and learnable parameters (even though more parameters do not necessarily mean more learning power). Here, we studied different architectures (Figure 3) that were all built upon the LSTM benchmark and shared nearly the same number of parameters (param. #). A “wider” LSTM was simply built by increasing *h*. A “deeper” LSTM was built by stacking two LSTM units as one unit. In particular, LSTM-MPS and LSTM-MERA were built and compared.

#### 4.1.1. Lorenz System

Figure 3a describes the forecasting task on the Lorenz system and shows the training results of the LSTM-based models. Δt=0.5 was chosen for discretization, which was large enough that the resultant time series hardly exhibited any pattern without the help of a phase line [Figure 3a, input 1–8].

In general, non-tensorized LSTM models performed worse than tensorized LSTM models. After the number of free parameters increased from 332 (benchmark) to 668±28, both the “wider” and “deeper” LSTMs showed signs of overfitting. The “deeper” LSTM yielded lower RMSE than the “wider” LSTM, confirming the common sense that a deep NN is more suitable for generalization than a wide NN [45]. Both LSTM-MPS and LSTM-MERA yielded better RMSE and showed no sign of overfitting. However, LSTM-MERA was more powerful, showing an improvement of ∼25% over LSTM-MPS in RMSE [Figure 3a].

#### 4.1.2. Logistic Map

Figure 3b describes a specific forecasting task on the simplest one-dimensional discrete-time map—the *logistic map*: predicting the target given only a three-step-behind input. Different LSTM models yielded very different results when learning this complex task. After the number of free parameters increased from 35 (benchmark) to 1142±89, all LSTM models yielded lower RMSE than the benchmark. Only LSTM-MERA was able to reach a much lower RMSE (presumably a global minimum) with a remarkable improvement of ∼94% over LSTM-MPS [Figure 3b]. We infer that the local minima reached by the other LSTM models might correspond to the infinite numbers of unstable quasi-periodic cycles in the chaotic phases. In fact, as shown in Figure 3b, Prediction 3, the benchmark fit the target better than LSTM-MERA for this specific example of a quasi-period-2 cycle. However, LSTM-MERA learned the full chaotic behavior and thus performed much better on general examples.

The learning process for the logistic map task was indeed very random, and different realizations yielded very different results. In many realizations, non-tensorized LSTM models did not even learn any patterns at all. By contrast, tensorized LSTM models were more stable in learning.

### 4.2. Comparison with Statistical/ML Models

We compared LSTM-MERA with more general models, including traditional statisical and ML models, including RNN-based architectures (Figure 4). Specifically, we looked into HOT-RNN/LSTM, which is also claimed to be able to learn chaotic dynamics (e.g., the Lorenz system) through tensorization [28]. Furthermore, for each model we fed its one-step-ahead predictions back so as to make predictions for the second step, and kept feeding back and so on. In theory, the prediction error at the *t*-th step should increase exponentially with *t* for chaotic dynamics [Equation (Equation 1)].

#### Gauss Iterated Map

We tested the one-step-ahead learning task on the *Gauss “cubed” map* on plain HO-RNN/LSTM [15] and its tensorized version HOT-RNN/LSTM [28]. The explicit “history” length was chosen to be equal to our physical Length *L*. The tensor-train ranks were all chosen to be equal to $D_{\mathrm{II}}$, as when we built the MPS structure in LSTM-MPS.

Figure 4 shows that neither HO-RNN nor HO-LSTM performed better than the benchmark, suggesting that introducing explicit non-Markovian dependence (Section B.1) is not helpful for capturing chaotic dynamics where the existing nonlinear complexity is never long-term. HOT-LSTM was better than the benchmark because of its MPS structure, suggesting that tensorization, on the other hand, is indeed helpful for forecasting chaos. LSTM-MERA was still the best, with an improvement of ∼88% over the benchmark. Interestingly, the benchmark itself as a vanilla LSTM was already much better than plain RNN architectures (HO-/HOT-RNN).

The learning task was next tested on fully connected deep NN architectures of depth ≤8 (equal to the input steps). At each depth three units were connected in series: a linear layer, a scaled exponential linear unit, and a dropout layer. Hyperparameters were determined by means of an optimal search. The best model having the lowest validation loss consisted of 17,950 free parameters. The task was also tested on GBT of maximum depth =8, as well as on the ARMA family (ARMA, ARIMA, FARIMA, and SARIMA), among which the best statistical model selected by Kalman filtering was ARMA(3,4).

With enough parameters, the deep NN became the second best (Figure 4). All RMSE values increased when making longer-step-ahead predictions, and for the four-step-ahead task the deep NN and LSTM-MERA were the only models that did not overfit and still performed better than the statistical model, ARMA, which made no learning progress but only trivial predictions.

### 4.3. Comparison with LSTM-MERA Alternatives

Here we tested the ability of LSTM-MERA in the learning of short-term nonlinear complexity by changing its NN topology (Figure 5). We expected to see that, to achieve the best performance, our tensorization (dashed rectangle in Figure 1) should indeed act on the state propagation path $c_{t}\to s_{t}$, not on $s_{t-1}\to s_{t}$ or $c_{t-1}\to c_{t}$.

#### Thomas’ Cyclically Symmetric System

We investigated different LSTM-MERA alternatives on Thomas’ cyclically symmetric system (Figure 5) in order to see if the short-term complexity could still be efficiently learned. The embedded layers, in addition to being located at Site A (the proper NN topology of LSTM-MERA), were also located alternatively at Site B, C or D for comparison. The benchmark was a vanilla LSTM with no embedded layers.

As expected, the lowest RMSE was produced by the proper LSTM-MERA but not its alternatives (Figure 5). The improvement of the proper LSTM-MERA over the benchmark was ∼60%. Interestingly, two alternatives (Site B, Site C) performed *barely* better than the benchmark even with more free learnable parameters. In fact, in the case in which the state propagation path $c_{t-1}\to c_{t}$ is tensorized (Site B), the long-term gradient propagation along cell states is interfered and the performance of LSTM is deterred; when the path $s_{t-1}\to s_{t}$ is tensorized (Site C), the improvement is the same as just on a plain RNN and is thus also limited. Hence, proper LSTM-MERA NN topology is critical for improving the performance of learning short-term complexity.

### 4.4. Generalization and Parameter Dependence of LSTM-MERA

The inherent advantage of LSTM-MERA and its ability to learn chaos have been shown. Hereafter investigated are its parameter dependence, as well as its generalization ability (Figure 6). Each following model (benchmark versus LSTM-MERA) was sufficiently trained through the same number of epochs so that it could reach the lowest stable RMSE. In-between check points were chosen during training, in which models were tested a posteriori on the test data to confirm that an RMSE minimum had eventually been reached.

#### 4.4.1. Rössler System

In theory, a chaotic time series of larger Δt should be harder to learn [Equation (Equation 1)]. This is confirmed in Figure 6a, in which a larger Δt corresponds to a larger RMSE for both models. The greatest improvement of LSTM-MERA over the benchmark was ∼76%, observed at Δt=5. The improvement was less when Δt increased, possibly because the time series became too random to preserve any feasible pattern even for LSTM-MERA. The improvement was also less when Δt was small, as the time series was smooth enough and the first-order (linear) time-dependence predominated, which a vanilla LSTM could also learn.

#### 4.4.2. Hénon Map

In view of the fact that the time-dependence is second-order [Figure 6b], there was no explicit and exact dependence between the input and target in the time series dataset. Different input steps were chosen for comparison. When input steps =1, there was no sufficient information to be learned other than a linear dependence between the input and target, and thus both the benchmark and LSTM-MERA performed the same [Figure 6b]. When input steps >1, however, the time-dependence could be learned implicitly and “bidirectionally” given enough history in length. LSTM-MERA constantly exhibited an average improvement of 45.3%, the fluctuation of which was mostly due to the learning instability not of LSTM-MERA but of the benchmark.

#### 4.4.3. Duffing Oscillator System

Based on Figure 6d, it was clearly observed that larger *L* yielded better RMSE values. The improvement related to *L* was significant. This result is not unexpected, since *L* determines the depth of the MERA structure, with a larger depth corresponding to better generalization ability.

#### 4.4.4. Chirikov Standard Map

As Figure 6d shows, by choosing different *P*, the greatest improvement of LSTM-MERA over the benchmark was ∼56%, observed at P=8. In general, there was no strong dependence on *P*.

#### 4.4.5. Real-World Data: Weather Forecasting

The advantage of LSTM-MERA was also tested on real-world weather forecasting tasks [Figure 6e,f]. Unlike for the synthetic time series, here we removed the first-layer translational symmetry [Equation (Equation 10)] previously imposed on LSTM-MERA so that presumed non-stationarity in real-world time series could be better addressed. To perform practical multi-step forecasting, we kept the one-step-ahead prediction architecture of LSTM, yet regrouped the original time series by choosing different prediction window lengths (Section C.3).

The improvement of LSTM-MERA over the benchmark was less significant. The average improvement was ∼3.0%, whereas the greatest improvement was ∼6.3%, considering that the prediction window length was small and reflecting that LSTM-MERA is better at capturing short-term nonlinear complexity rather than long-term non-Markovianity. Note that, in the second dataset [Figure 6f], we deliberately used a very small number (=128) of training data to test the overfitting resistibility of the models. Interestingly, LSTM-MERA did not generally perform worse than vanilla LSTM even with more parameters, probably due to the deep architecture of LSTM-MERA.

## 5. Discussion and Conclusions

The limitations of our model mostly come from the fact that it is only better than traditional LSTM at capturing short-term nonlinearity but not long-term non-Markovianity, and thus its improvement on long-term tasks such as sequence prediction would be limited. That being said, the advantages of tensorizing state propagation in LSTM are evident, including: (1) Tensorization is the most suitable method for the forecasting of nonlinear chaos since nonlinear terms are treated explicitly and weighted equally by polynomials. (2) Theoretical analysis is conductible since an orthogonal polynomial basis on *k*-Sobolev space is always available. (3) Tensor decomposition techniques (in particular, from quantum physics) are applicable, which in turn can identify chaos from a different perspective, i.e., tensor complexity (tensor ranks, entropies, etc.).

Our tensorized LSTM model not only offers a general and efficient approach for capturing chaos—as demonstrated by both theoretical analysis and experimental results, showing great potential in unraveling real-world time series—but also brings out a fundamental question of how tensor complexity is related to the learnability of chaos. Our conjecture that a tensor complexity of $S_{\alpha }\left(l\right)=\Theta \left(\log l\right)$ in terms of α-Rényi entropy [Equation (Equation 6)] generally performs better than $S_{\alpha }\left(l\right)=\Theta \left(1\right)$ in chaotic time series forecasting will be further investigated and formalized in the near future. 

## Figures and Tables

**Figure 1 entropy-23-01491-f001:**
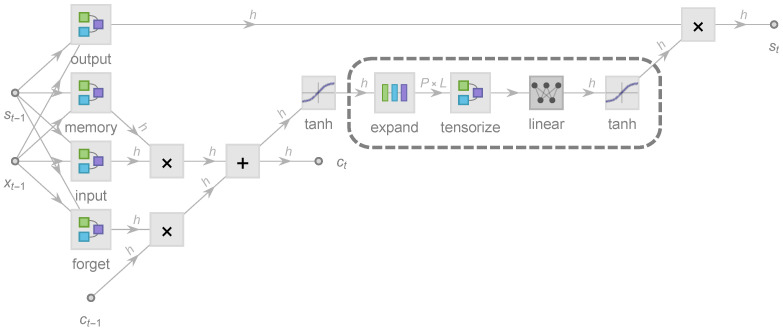
Architecture of a long short-term memory (LSTM) unit in the most common form of four gates: input (*i*), memory (*m*), forget (*f*), and output (*o*), enhanced by tensorized state propagation with four additional layers embedded (dashed rectangle): *expand* [Equation (Equation 4)], *tensorize* (Figure 2), *linear*, and a *tanh* activation function. *d* is the input dimension of $x_{t}$, and *h* is the hidden dimension of state $s_{t}$ and cell state $c_{t}$. An *h*-dimensional vector $\tanh c_{t}$ is first expanded into a P×L-dimensional matrix where *L* and *P* are dubbed the physical length and physical degrees of freedom (DOF), respectively. Then, the matrix is tensorized into an *L*-rank tensor of dimension $P^{L}$ and passed forward. The effectiveness of this architecture is investigated in Section 4.3.

**Figure 2 entropy-23-01491-f002:**
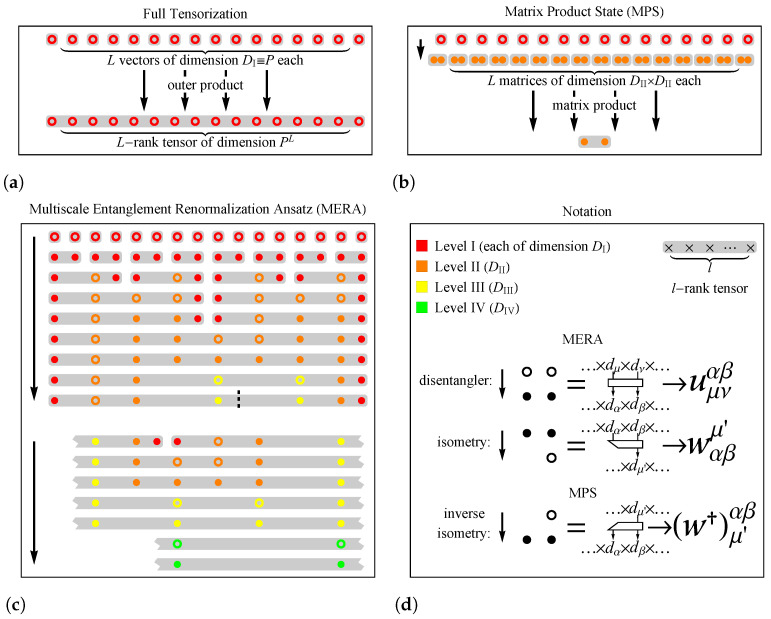
Tensorize layer: quantum entanglement structures. (**a**) Full tensorization. (**b**) Matrix product state (MPS). (**c**) Multiscale entanglement renormalization ansatz (MERA). The MPS and MERA are tensor representations that are widely used for characterizing many-body quantum entanglement in condensed matter physics. (**d**) Notations. A full tensor can be represented by introducing multiple auxiliary and learnable tensors (e.g., disentanglers and isometries, as used in MERA, and inverse isometries, as used in MPS) of different virtual dimensions $\left\{D_{I},D_{\mathrm{II}},\cdots \right\}$ labeled by different levels, rendered in different colors. The first-level virtual dimension is $D_{I}\equiv P$, the physical DOF by definition. Other virtual dimensions $\left\{D_{\mathrm{II}},\cdots \right\}$ are free hyperparameters to be chosen, the larger of which should better represent the full tensor. The numbers of applicable levels in (**a**,**b**) are always constant (one and two, respectively), yet the number of applicable levels in **c** is $\log_{2}L$, depending on the physical length *L*.

**Figure 3 entropy-23-01491-f003:**
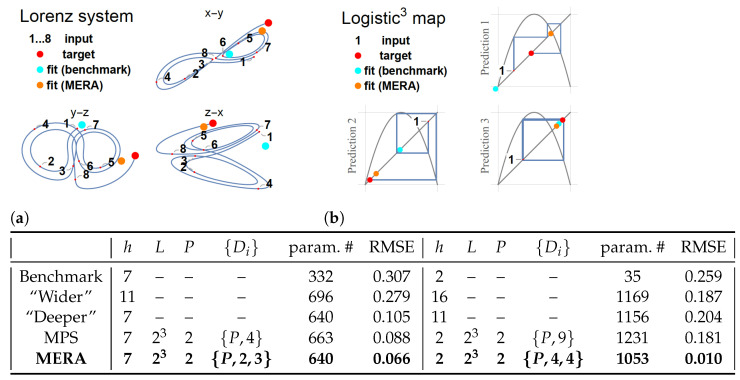
Comparison of different LSTM-based architectures. (**a**) The Lorenz system is a three-dimensional continuous-time dynamical system notable for its chaotic behavior. Discretization: Δt=0.5. Input steps =8, training:validation:test=2400:600:2000, and number of epochs =120 for all models. (**b**) Logistic “cubed” map, i.e., a logistic map re-sampled every three steps. Input steps =1, training:validation:test=8000:2000:500, and number of epochs =200 for all models. Note that unlike continuous-time dynamical systems, chaos in discrete maps is generally harder to learn.

**Figure 4 entropy-23-01491-f004:**
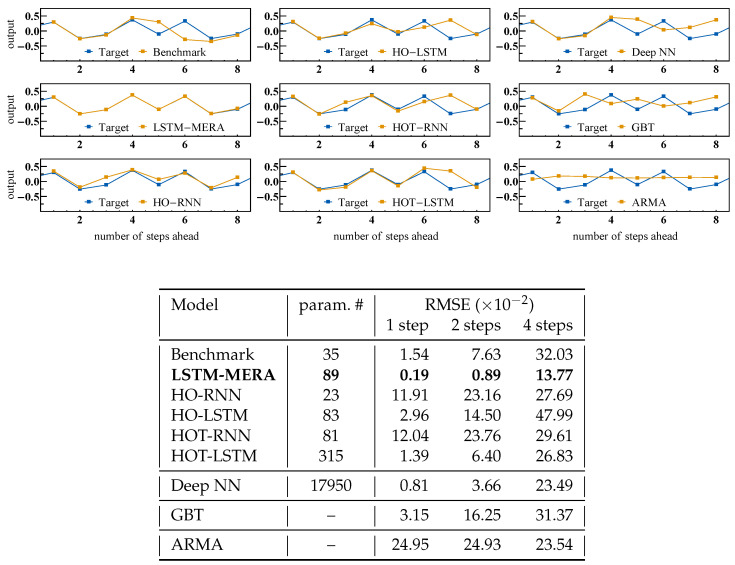
Comparison of different statistical/ML models on Gauss “cubed” map, i.e., a Gauss iterated map re-sampled every three steps. Note that the Gauss iterated map is a one-dimensional chaotic map of which the dynamics is smoother than the logistic map and should be easier to learn. Input steps =8. For RNN-based models, h=2, $L=2^{2}$, P=2, $\left\{D_{I},D_{\mathrm{II}},\cdots \right\}=\left\{P,2\right\}$, training:validation:test=8000:2000:500, and number of epochs =200. The explicit “history” length used in HO-RNN/LSTM [15] and HOT-RNN/LSTM [28] is also *L*, and the tensor-train ranks are all $D_{\mathrm{II}}$. Deep NN: depth =8 (= input steps). GBT: maximum depth =8. ARMA family: ARMA(3,4).

**Figure 5 entropy-23-01491-f005:**
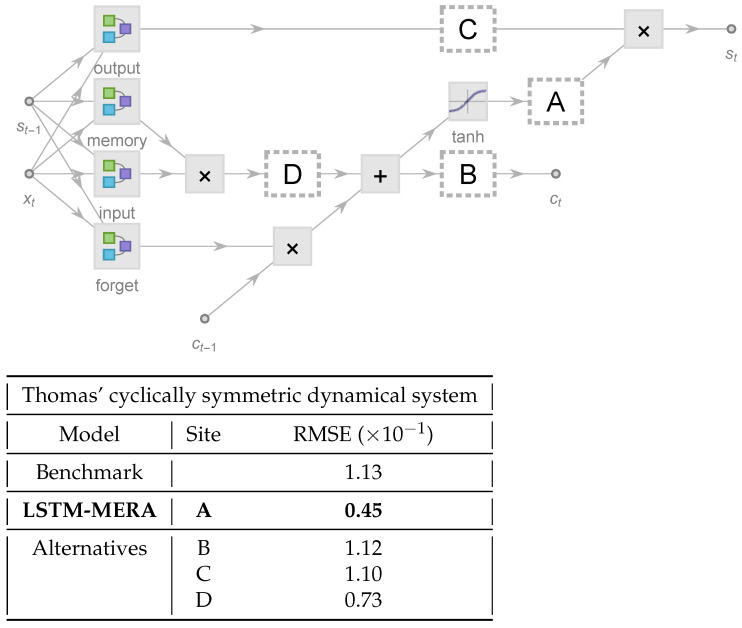
Comparison of LSTM-MERA (where the additional layers from Figure 1 are located at Site A) with its alternatives (where the additional layers are instead located at Sites B, C or D), tested on Thomas’ cyclically symmetric system, a three-dimensional chaotic dynamical system known for its cyclic symmetry Z/3Z under change of axes. Discretization: Δt=1.0. Input steps =8, h=4, $L=2^{4}$, P=4, $\left\{D_{I},D_{\mathrm{II}},\cdots \right\}=\left\{P,2,2,4\right\}$, training:validation:test=2400:600:2000, and number of epochs =40.

**Figure 6 entropy-23-01491-f006:**
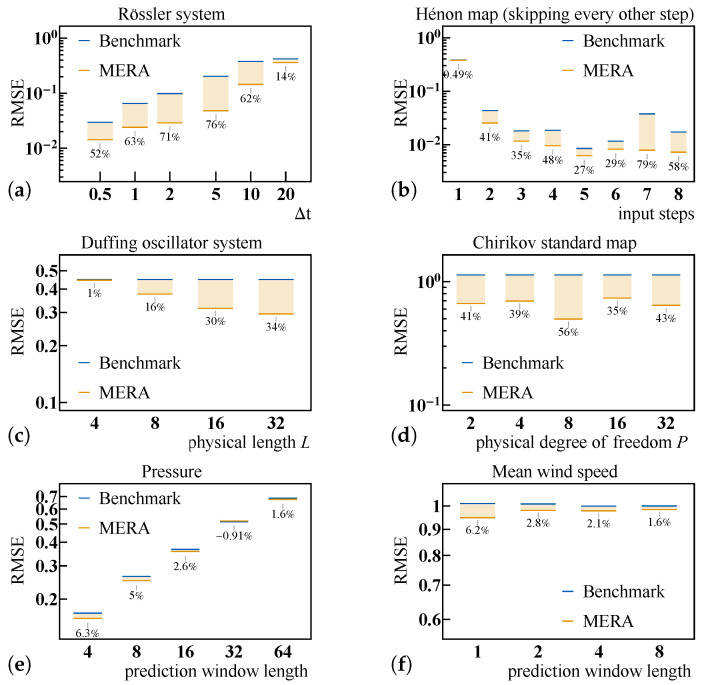
Generalization and parameter dependence of LSTM-MERA. (**a**) Rössler system, another three-dimensional chaotic dynamical system similar to the Lorenz system. Discretization: varying Δt. Input steps =4, h=4, $L=2^{4}$, P=2, and $\left\{D_{I},D_{\mathrm{II}},\cdots \right\}=\left\{P,2,2,4\right\}$. (**b**) One-dimensional, second-order Hénon map, re-sampled by skipping every other step. h=4, $L=2^{3}$, P=2, and $\left\{D_{I},D_{\mathrm{II}},\cdots \right\}=\left\{P,2,4\right\}$, whereas the input steps varied. (**c**) Duffing oscillator system. Discretization: Δt=10.0. Input steps =8, h=4, P=4, and $\left\{D_{I},D_{\mathrm{II}},\cdots \right\}=\left\{P,3,3,\cdots \right\}$ of which the length varies with *L*. (**d**) Chirikov standard map. Input steps =2, h=2, $L=2^{3}$, and $\left\{D_{I},D_{\mathrm{II}},\cdots \right\}=\left\{P,4,4\right\}$ where *P* varies. (**e**) Pressure, sampled every eight minutes. Input steps =16, h=4, $L=2^{4}$, P=4, $\left\{D_{I},D_{\mathrm{II}},\cdots \right\}=\left\{P,2,2,4\right\}$, and $\mathrm{training}:\mathrm{validation}:\mathrm{test}=6400:1600:\left(≳13,800\right)$. (**f**) Mean wind speed, daily sampled. Input steps =16, h=4, $L=2^{3}$, P=4, $\left\{D_{I},D_{\mathrm{II}},\cdots \right\}=\left\{P,2,2\right\}$, and $\mathrm{training}:\mathrm{validation}:\mathrm{test}=128:32:\left(≳7000\right)$.

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
