# Peer review of "Entanglement-Structured LSTM Boosts Chaotic Time Series Forecasting"

_entropy, 2021, doi:10.3390/e23111491_

Round 1

Reviewer 1 Report

Authors provide a LSTM architecture implemented using many-body structures. They use the underlying physical motivation to conjecture that the ability to learn in LSTM depends on the complexity of the tensor decomposition. In my opinion, the presented results are interesting and might be published after introduction of minor changes (see below). The overall quality of the paper is high, and the presented results are supported by the relevant source code.

Minor comments:

1) I can recommend including some citation relevant for the current research. In particular, analysis presented in

Multivariate cumulants in outlier detection for financial data analysis,Physica A, 124995 by   K.Domino, 

as well as 

The use of the Hurst exponent to predict changes in trends on the Warsaw Stock Exchange
Physica A: Statistical Mechanics and its Applications 390 (1), 98-109

by the same author, could be relevant from the perspective of chaotic behavior.

2) I suggest improving figure A1. as it is hardly readable.

Author Response

Thanks a lot for the time devoted to reviewing our manuscript, associated with professional suggestions. We have carefully modified the manuscript accordingly align with your helpful comments:

  1. We have added citation to both works mentioned in the comments, both of which, indeed, discuss closely relevant application scenarios of nonlinear time series modeling.
  2. We have updated the Fig. A1 to a newer version, which now shows more details of the data.

Thanks again for your useful advice!

Reviewer 2 Report

I think this paper is a nice effort toward the topic discussed. The authors are just invited to discuss more deeply how well (or not) the forecast techniques could predict (in general) suitable (or non-suitable) results. The paper can be updated a little bit even at the level of English and style. Also, the paper can be split in a more impressive way than this: a reader has problems to understand the structure proposed by the authors that are invited to re-better arrange the various sections and subsections. Besides these small points, the paper can be accepted for publication. 

Author Response

Dear reviewer,

Thanks for your time devoted to the manuscript review associated with extremely useful suggestions. We have carefully revised the manuscript based on your advice:

  1. "the paper can be split in a more impressive way than this... to re-better arrange the various sections and subsections."
    We have split and re-organized several sections/subsections, for a better delivery of both ideas, intuitions, and rigorous analysis;
  2. "The authors are just invited to discuss more deeply how well (or not) the forecast techniques could predict (in general) suitable (or non-suitable) results."
    We have elaborated more one some proposed methodologies, e.g. the entanglement entropy measure of tensor complexity and model capacity, together with a connection to their original usage in related field, e.g. condensed matter theories and quantum information.
  3. "The paper can be updated a little bit even at the level of English and style."
    We have further checked and corrected some grammar issues and typos within the previous version.